# Impact of Vitamin D Status and Nutrition on the Occurrence of Long Bone Fractures Due to Falls in Elderly Subjects in the Vojvodina Region of Serbia

**DOI:** 10.3390/nu16162702

**Published:** 2024-08-14

**Authors:** Nemanja Gvozdenović, Ivana Šarac, Andrijana Ćorić, Saša Karan, Stanislava Nikolić, Isidora Ždrale, Jelena Milešević

**Affiliations:** 1Faculty of Medicine, University of Novi Sad, 21137 Novi Sad, Serbia; nemanja.gvozdenovic@mf.uns.ac.rs (N.G.); 912007d23@mf.uns.ac.rs (A.Ć.); 1296d18@mf.uns.ac.rs (S.K.); stanislava.nikolic@mf.uns.ac.rs (S.N.); 015169@mf.uns.ac.rs (I.Ž.); 2Clinic for Orthopedic Surgery and Traumatology, University Clinical Center of Vojvodina, 21137 Novi Sad, Serbia; 3Center of Research Excellence in Nutrition and Metabolism, Institute for Medical Research, National Institute of Republic of Serbia, University of Belgrade, 11129 Belgrade, Serbia; ivana.sarac@imi.bg.ac.rs; 4Center of Laboratory Medicine, University Clinical Center of Vojvodina, 21137 Novi Sad, Serbia

**Keywords:** fracture, elderly population, vitamin D, diet, frailty

## Abstract

Bone fractures are a significant public health issue among elderly subjects. This study examines the impact of diet and vitamin D status on the risk of long bone fractures due to falls in elderly subjects in Vojvodina, Serbia. Conducted at the University Clinical Center of Vojvodina in autumn/winter 2022–2023, the study included 210 subjects >65 years: 105 (F: 80/M: 15) with long bone fractures due to falls and 105 (F: 80/M: 15) controls. Groups were similar regarding age and BMI. Dietary intakes (by two 24-h recalls) and serum vitamin D levels were analyzed. The fracture group had a significantly lower median daily vitamin D intake (1.4 μg/day vs. 5.8 μg/day), intake of calcium, energy, proteins, fats, fibers, dairy products, eggs, fish, edible fats/oils, and a higher intake of sweets (*p* < 0.001 for all). Serum vitamin D levels were significantly lower in the fracture group (40.0 nmol/L vs. 76.0 nmol/L, *p* < 0.001). Logistic regression identified serum vitamin D as the most important protective factor against fractures, and ROC curve analysis indicated that serum vitamin D levels > 50.5 nmol/L decreased fracture risk. Nutritional improvements (increased intake of vitamin D and protein sources such as fish, eggs, and dairy), increased sun exposure, and routine vitamin D supplementation during winter are advised.

## 1. Introduction

Bone fractures are a global public health problem, as they often lead to absenteeism, reduced productivity, disability, impaired quality of life, and increased risk of death, particularly among the elderly population. Such injuries are a significant burden on individuals, families, societies, and healthcare systems [1]. The incidence of long bone fractures increases every year—with the extension of life expectancy, followed by an increasing number of comorbidities, more frequent traffic traumatism, and lifestyle changes [2]. Fractures of long bones can occur due to the action of the force of high intensity in younger individuals, as well as under the action of the force of small intensity in the elderly population [3]. Falls are the most common cause of bone fractures in the elderly, and they occur associated with conditions such as osteopenia, osteoporosis, sarcopenia, and neuromotor degradation [4]. In elderly patients (over 65 years of age), ground-level fall is considered a major cause of injury and admission to hospital facilities around the world [5]. The Centers for Disease Control and Prevention suggests that fractures are classified among the 20 most common diagnoses that were treated within emergency medical services. The most common fractures due to bone insufficiency (osteoporosis) include fractures of the vertebrae, fractures of the long bones—the lower end of the sternum, the upper end of the femur, the distal end of the radius, the trunk, and shoulder bone [4,6,7]. With the increase and aging of the population, the incidence of osteoporotic fractures increases [1]. The risk of osteoporotic fractures among the older population in developing countries (including Serbia) has been increasing in recent years, due to lifestyle changes that favor osteoporosis [2,8]. It has been estimated that every second woman after the age of 65 will have at least one fracture for the rest of her life. One-third of elderly patients experience trauma every couple of weeks, of which 20% result in a fracture [9]. In Serbia, the estimated annual incidence of hip fractures due to falls among the population of 70 and more years old is about 20% [1,2]. Treatment of such patients is usually followed by complications; it is time-consuming and economically demanding, due to the cost of surgery, possible rehospitalization, and rehabilitation of patients, and often is associated with life-threatening conditions [3,9,10,11,12]. For example, some data suggest a mortality rate of 17% to 25% within 1 year following hip fractures or surgery in older adults, and the overall mortality rate was estimated to be ~20% [13,14]. In a recent study in the USA, in older adults aged 60–89 years, hip fractures increased 1-year mortality rates by 5–7% in females and 6–11% in males, while among females and males 60–69 years old, even 5.4-fold and 4-fold greater 1-year age-adjusted mortality risk was shown following hip fracture compared to that estimated for the general population [15]. Other long bone fractures among adults 60–89 years of age were also associated with an increased mortality risk [15].

Investigating interventional risk factors for osteoporotic fractures in the elderly, such as body mass index (BMI), type of diet, and physical activity is crucial for a better understanding of the mechanism of fractures. Low serum micronutrient values (such as levels of calcium, magnesium, copper, iron, and vitamins D and K), as well as protein deficiency, and low dietary fiber intake, are considered important risk factors associated with the incidence of osteoporotic fractures in the elderly [16,17,18,19,20]. Additional risk factors for fractures include a low body mass index (BMI) and a tendency to fall, because of an individual’s decreased balance, musculoskeletal instability, visual impairments, hypotension, and medications [21].

The importance of proper nutrition and its positive impact on endochondral osteogenesis after trauma is often addressed in medical literature [22]. Malnutrition (both hidden hunger and visible protein/energy malnutrition) is of great interest to researchers, as it is one of the modifiable risk factors related to fractures [23,24,25,26]. Elderly patients with fractures are often malnourished and in constant struggle to maintain adequate nutrition. Rates of malnutrition in patients with trauma of the musculoskeletal system are known to vary between 10% and 50% [25]. Adequate nutrition is one of the therapeutic strategies in post-traumatic recovery, as bone homeostasis depends on the interaction of several minerals and vitamins, such as calcium and vitamin D. Although several studies on long bone fractures in animals have shown that diet optimization leads to improved fracture healing, information about human nutrition status and outcomes after orthopedic trauma is scarce [27].

Bone is a metabolically active tissue that quickly responds to changes in nutritional status. It consists of ~35% protein, mainly collagen. Minerals make up the remaining ~65% of bone tissue, predominantly calcium and phosphate [28]. Vitamin D is a liposoluble vitamin and promotes the deposition of calcium into the bone. It has both endogenous (D3) and exogenous (D2 and D3) sources. In the skin, provitamin D3 (7-dehydrocholesterol, 7-DHC) is, by ultraviolet B (UV-B) radiation, converted into previtamin D3, further converted by thermal isomerization into vitamin D3, which is then activated into 25-hydroxy vitamin D3 in the liver and, sequentially, 1,25 dihydroxy vitamin D3 in the kidneys. The synthesis of vitamin D3 through sunlight is highly correlated with the seasons, meaning that vitamin D synthesis is greatly reduced or does not occur in the autumn and winter months [29].

The risk groups for poor vitamin D status include elderly people, people with kidney or gastrointestinal tract diseases, people who do not expose themselves to a sufficient amount of UV light, and those who use medications that adversely affect the concentration of vitamin D. The adequate vitamin D status, defined by serum 25(OH)D > 50 nmol/L, is present in less than 50% of the world’s population, especially in autumn and winter [30,31]. In Serbia, there is particularly a high prevalence of vitamin D deficiency, affecting roughly more than two-thirds (68.5%) of subjects [32]. Prevention strategies include adequate sun exposure, regular fish consumption (1–2 times per week), fortification of foods with vitamin D, and the use of vitamin D supplements [30].

The objective of this work was to examine the impact of diet on the occurrence of long bone fractures due to falls in elderly subjects, with an emphasis on the effect of serum vitamin D levels, as well as micronutrient, macronutrient, and specific food groups’ intake. We hypothesized that there would be additive/synergistic effects of different dietary and lifestyle factors on the increased or decreased risk for long bone fractures among elderly subjects. Additionally, the study examined the dietary characteristics and status of vitamin D of the older population, with or without bone fractures, in a particular region of Serbia, data on which are lacking.

## 2. Materials and Methods

### 2.1. Ethical Considerations and Study Participants

The research was approved by the Ethics Committee of the Clinical Centre of Vojvodina in Novi Sad (decision No. 00-249 from 29 December 2022). The subject’s consent to participate in the study was obtained in accordance with the Helsinki Declaration.

The study was conducted as a case-control study at the Clinic for Orthopedic Surgery and Traumatology of the University Clinical Center of Vojvodina, Novi Sad, Serbia, in the period autumn/winter 2022–2023. All the subjects were residents of the Vojvodina region of Serbia (latitude between 41°53’ and 46°11’ N). This case-control study initially included 240 elderly subjects over 65 years of age (120 cases and 120 controls from the same geographical region, evenly balanced for sex, 90 women in each group), but after applying the exclusion criteria (described below), the final sample included 210 subjects: the case group consisted of 105 elderly subjects (females *n* = 80), with fractures of long bones due to a ground-level fall, who were treated at the Clinic for Orthopedic Surgery and Traumatology of the Clinical Center in the aforementioned period and satisfied exclusion criteria. The control group consisted of 105 elderly subjects (females *n* = 80), of the same region of residence (Vojvodina), who were recruited by a random phone call sampling (random digit dialing—RDD), matched by sex to the case group, who did not have a long bone fracture at the time of the study due to falls, and who satisfied inclusion/exclusion criteria. All data collection and laboratory analyses in the control group were performed as soon as possible after the requirement, to avoid the possible influence of seasonal changes.

Criteria for the inclusion of cases in the study:65 or over years of ageFracture of a long bone to a ground-level fallVoluntary consent of a patient to participate in the study

Criteria for the inclusion of controls in the study:65 or over years of ageNo fracture of a long bone to a ground-level fallVoluntary consent of a patient to participate in the study

Criteria for excluding participants from the study:Patients under the age of 65Refusal of the patient to participate in the studyInability to give written consentForeign citizensCommunication problemsDeath in the course of the studyVitamin D and calcium substitution therapy started less than 6 months agoExposure to the UV-B light 1 month before the start of the test (going to the sea, mountain, and solarium).Medical conditions that can influence the overall/bone health or vitamin D status (kidney diseases, malignant diseases, hematologic disorders, idiopathic hypercalciuria, diabetes mellitus type 1, parathyroid gland diseases, adrenal gland diseases, acromegaly, rheumatoid diseases, chronic gastrointestinal and liver disorders—inflammatory bowel diseases, celiac disease, gastrointestinal resection and bariatric surgery, cystic fibrosis, neurological disorders, long-term immobilization, etc.) [33].Long-term uses of medications connected with vitamin D, calcium, and bone metabolism (bisphosphonates, calcitonin, corticosteroids, antiepileptic drugs, SSRIs, thyroxin, TSH, gonadotropin-releasing hormone antagonists, progestins, tamoxifen, loop diuretics, aluminum-containing antacids, chemotherapy, and heparin) [34].Missing any presented data

### 2.2. Required Sample Size Calculation

Required sample size calculation, based on the estimated meaningful difference in serum vitamin D that could be associated with risk of fractures (effect size, ES, 20 nmol/L) and standard deviation of serum vitamin D levels (SD, 30 nmol/L) estimated from a pilot study in the same elderly population [35], using the formula: *n* (minimal per group) =  2(Zα + Zβ)2SD^2^/(ES)^2^, with Zα = 1.9600 and Zβ = 0.8416 for α  =  0.05 and β  =  0.2, i.e., at the level of significance of 5% and study power of 80% [36], was 72 subjects in total. However, for logistic regression models, we needed at least 200 subjects for a model that included 10 independent covariates [37]. Considering possible dropout and unfitting for the study, we increased the total number of recruited subjects to 240, which, after applying the exclusion criteria, was reduced to 210.

### 2.3. Socioeconomic, Medical, Lifestyle, Anthropometric, and Blood Pressure Data

Self-reported information on socio-demographic, medical (presence of chronic diseases, administration of therapy) characteristics of the subjects and their lifestyle (smoking, self-assessed physical activity level) was collected. Physical activity was self-estimated on the scale scoring from one (very inactive) to five (very active). Data on height and weight were obtained by standardized measurement methods [38] and used to calculate BMI [39]. Blood pressure was measured by a trained clinician using the standardized method [40].

### 2.4. Dietary Intakes

To determine the dietary intake of vitamin D, a twice-repeated 24-h dietary recall was conducted, which included collecting information about the complete food and beverage consumption of a patient from a previous day. The 24-h dietary recall was repeated after at least seven days in order to capture the variability of a diet. Besides, additional information on vitamin D supplement consumption was collected [41]. Each subject was interviewed by a phone call or in personal contact (the latter only with patients who were hospitalized after a fracture). The data from the surveys were processed using the DIET ASSESS & PLAN Advanced (DAP) software (https://www.deltaelectronic.net/dap/efsa/ishrana.php, accessed 22 June 2024) that allows assessment of dietary intake of nutrients at the daily, individual, and population level [42], by using the data from the Serbian food composition database on the energy, protein, carbohydrate, fat, calcium, and vitamin D content of food, as well as the contribution of certain food groups to total energy consumption [43]. Due to the lack of our national dietary recommendations, assessment of nutritional adequacy was performed by comparing with the US Dietary Guidelines for Americans, 2020–2025 (for energy and macronutrients) [44], and the USA Institute of Medicine (IOM), the Endocrine Society, and the European Food Safety Authority (EFSA) dietary recommendations (for vitamin D and calcium intakes) [45,46,47].

### 2.5. Determination of Vitamin D Status

The study involved determining the status of vitamin D, i.e., serum-25(OH)D. Blood samples were collected in the late summer/early autumn season (September 2022) and late autumn/early winter season (November 2022–January 2023) at the clinical laboratory of the University Clinical Center of Vojvodina, by a standard procedure, and immediately analyzed using the Chemiluminescent Microparticle Immuno Assay—CMIA, which is a standard technique used to measure vitamin D in clinical settings, due to its simplicity, time efficiency, and economic costs. It is often the method of choice for quantitative determination of vitamin D metabolites due to its sensitivity, accuracy, and repeatability. The main limitation of immunochemical methods is cross-reactivity of antibodies used in the test, which means that this method cannot distinguish between D2 and D3 forms of metabolites 25(OH)D or other inactive vitamin D metabolites [48]. The samples were analyzed on the Abbott Alinity i (Abbott Diagnostics, Lake Forest, IL, USA), using the commercial Architect 25-OH D Reagent kit. Quality control within-run and between-run imprecision (expressed as coefficient of variance, CV) was, respectively, 2.8–3.6% and 3.3–4.8%, depending on whether the control was of lower, medium, or higher range. Assessment of vitamin D nutritional status adequacy was performed by comparing with the IOM recommendations (from 2011), the EFSA recommendations (from 2016), the International Osteoporosis Foundation (IOF) recommendations (from 2010), the Endocrine Society clinical practice guidelines (from 2011), and the American Geriatrics Society recommendations (from 2014) [45,46,47,49,50,51].

### 2.6. Statistics

The statistical analyses were carried out using the IBM SPSS 25 Statistics statistical package, and the two-tailed statistical significance was determined at the level *p* < 0.05. Distribution assessment of the numeric continuous data was conducted via the Kolmogorov–Smirnov test, the skewness and kurtosis measures, standard errors, and a visual inspection of histograms, Q-Q plots, and box plots. Descriptive statistics was presented as the mean and standard deviation (SD), median and interquartile range (IQR), minimum and maximum values, or absolute frequency of occurrence (*n*) and percentages (%), depending on the nature of the variable. The groups we compared (depending on the data distribution) by using the Student *t*-test or the Mann–Whitney U test (in case no adjustment was performed) and parametric or non-parametric one-way analysis of covariance (ANCOVA) (in case of adjustment for age, sex, BMI, smoking, physical activity level, and/or season). The disparities in frequencies were tested through Pearson’s chi-square or Fisher’s exact test. Correlations without and with adjustments (for age, sex, BMI, smoking, physical activity level, and/or season) were tested by parametric Pearson’s correlation and partial Pearson’s correlation or non-parametric Spearman’s rank correlation and partial Spearman’s rank correlation. The risk for fractures was determined by logistic regression, in a univariate model and enter/stepwise multivariate models (with included covariates that satisfied the univariate models and did not show multicollinearity, tested by the variance inflation factors, VIFs). Receiver operating characteristic (ROC) curve analysis was performed to select the best cutoff value of serum vitamin D levels, under which the risk for fractures significantly increases [52,53].

## 3. Results

### 3.1. General Data

The proportion of women was the same in both groups, more than three-thirds (76.2%). The age of subjects varied from 65 to 87 years. The groups were slightly different in terms of age and BMI, with subjects with fractures being only slightly older (*p* = 0.044) and with higher BMI (*p* = 0.044) (Table 1). In the case group, more than half belonged to the overweight category and about one-third belonged to normal weight, while in the control group, the proportion of normal weight and overweight subjects was quite equal. Negligible proportion of participants was obese in both groups, without statistical difference. However, the level of physical activity was significantly different, with most subjects being more active than inactive (moderately and highly active) in the control group, while in the group with fractures, the majority was neutral or more/less inactive (*p* < 0.001). At the same time, the proportion of those who previously reported any kind of fractures was significantly higher in the case group (66.7% vs. 16.2%, *p* < 0.001). Additionally, in the group with fractures, the proportion of active smokers and former smokers was much higher, as well as the proportion of smokers who smoked more than 10 cigarettes per day (*p* < 0.001). Regarding the education level, marital status, and number of household persons, the groups were not different (Table 1).

The most frequently reported diseases were hypertension, diabetes type 2, coronary heart disease, osteoporosis, arthritis, gastritis, cataracts, glaucoma, mental diseases (anxiety, depression, and dementia), and HOPB (asthma and chronic bronchitis). The most frequently reported used drugs were antihypertensive and glucose-regulatory agents. The case group reported more chronic diseases, particularly circulatory, endocrine, respiratory, and mental diseases. The systolic blood pressure was significantly higher in the case group (*p* < 0.001). No significant differences were present in terms of neoplasms, hematological, immunological, neurological, digestive, musculoskeletal, skin, and genitourinary diseases.

The season of examination and blood sampling was late summer/early autumn (September) in the majority of subjects in both groups, while the others were examined in the late autumn-early winter (November–January), without statistical difference between the two groups (Table 1).

### 3.2. The Dietary Intake of Energy and Macronutrients

In the group with fractures, the total caloric intake was much lower than in the control group, due to a lower intake of protein and fats, while the intake of carbohydrates was not different. This resulted in a significantly lower proportion of protein and fats in the dietary energy intake, while the proportion of carbohydrates was much higher. Median (IQR) intake of protein per kilogram body weight was 0.56 (0.46–0.77) g/kg/day vs. 0.99 (0.74–1.21) g/kg/day (Mann–Whitney test, *p* < 0.001). In the case group, 77.1% had a protein intake lower than 0.8 g/kg/day [54], and 83.8% lower than 1.0 g/kg/day [55]. In contrast, in the control group, 70.5% had a protein intake higher than 0.8 g/kg/day, and 48.6% higher than 1.0 g/kg/day (Pearson chi-square test: *p* < 0.001 for both). The intake and proportion of alcohol were just slightly higher in the case group, but only a small proportion of the subjects consumed alcohol in both groups (six subjects in the case group and two subjects in the control group consumed at least one unit of alcohol, *p* = 0.280). The dietary fiber intake was also lower in the case group (Table 2). All these differences in macronutrient intakes remained significant even when the corrections for sex, age, BMI, smoking status, physical activity level, and season were taken into consideration (non-parametric ANCOVA, *p* < 0.001 for all except for alcohol, *p* = 0.017, and percent alcohol in the energy intake, *p* = 0.014).

In the whole sample, total energy, protein, and fat intakes showed only a weak correlation with BMI (Spearman’s correlation coefficients r_s_ = 0.215, *p* = 0.002, r_s_ = 0.201, *p* = 0.003, r_s_ = 0.180, *p* = 0.009, respectively).

When compared with the current US dietary guidelines (2020–2025) for the intake of energy and percentage of energy coming from different macronutrients [44], 87.6% of the cases and 40.0% of the controls had a total energy intake below recommendations (1600 kcal/day for females and 2000 kcal/day for males), which was significantly different (*p* < 0.001), while 29.5% of the cases and 10.5% of the controls had an energy intake even below 1200 kcal/day (*p* = 0.001). At the same time, in the whole sample, a proportion of those with fat contribution in the diet above the recommended upper limit (35% of energy) was very high, 61.9% of the cases and 88.6% of the controls (*p* < 0.001), and 40.0% of the cases and 74.3% of the controls had fat contribution even above 40% (*p* < 0.001). In 22.9% of cases and only two controls the contribution of proteins was below 10% of energy (*p* < 0.001), others had protein contribution 10–35%, while the insufficient contribution of carbohydrates in total energy intake (below 45%) was present in 35.2% of the cases and even 84.8% of the controls (*p* < 0.001) (Appendix A).

### 3.3. The Intake of Different Food Groups

In the group with fractures, the number of calories coming from dairy products, eggs, fish, added fats, and oils was significantly lower than in the control group (*p* < 0.001 for all), the number of calories coming from sugars was significantly higher (*p* < 0.001), while the proportion of calories coming from cereals, meat, vegetables, fruits, nuts, and alcoholic and non-alcoholic beverages was not different, compared with the control group (Table 3). All these differences in food group consumption remained significant even after the corrections for sex, age, BMI, smoking status, physical activity level, and season (non-parametric ANCOVA, *p* < 0.001 for all except for eggs, *p* = 0.003). In both groups, the highest proportion of calories was coming from cereals and added fats or oils (making together about 48% of energy in both groups), then meat or dairy products (making together about 17 and 23% of energy, respectively, in the case and control group). Offal (liver and liver pate) was rarely eaten, by only eight subjects with fractures and four control subjects, in small amounts, and on average contributed daily with only 15.5 Kcal and 9.0 Kcal, respectively, in the case and control group. Fish was almost not eaten in the case group (only one subject reported), but in the control group it made about 5.5% of energy. Vegetables and fruits together made about 9% and 7% of energy, while sweets and beverages together made about 8% and 3% of energy, respectively, in the case and control groups (Appendix A).

### 3.4. The Intake of Vitamin D and Calcium through Food and Supplements

It was found that the average daily intake of vitamin D only through diet, as shown in Figure 1 and Appendix A, differed statistically significantly (*p* < 0.001, Mann–Whitney test) between the case group (median: 1.4, IQR: 0.9–2.7 μg/day) and the control group (median: 5.8, IQR: 3.3–8.6 μg/day). The difference in vitamin D dietary intake remained significant even after the corrections for sex, age, BMI, smoking status, physical activity level, and season (non-parametric ANCOVA, *p* < 0.001). In the case group, the median intake was higher in the late autumn/early winter season, compared with the late summer/early autumn season, 1.2 (0.7–1.7) μg/day vs. 2.6 (1.3–4.8) μg/day, while in the control group it was the opposite: 6.1 (3.7–9.6) μg/day vs. 4.5 (1.9–7.8) μg/day (Figure 2).

It is important to emphasize that none of the groups on average meet the daily needs for this vitamin through the usual diet, and the majority of respondents in both groups do not meet their daily needs. In both groups the proportion of those with an inadequate intake of vitamin D according to the European Food Safety Authority (EFSA) recommendations for subjects above 65 years old (<15 μg/day), was high: 102 subjects (97.1%) in the case group vs. 90 subjects (85.7%) in the control group (Fisher’s exact test, *p* = 0.005), while the proportion of those with an inadequate intake of vitamin D according to the IOM recommendations for those above 70 years old (<20 μg/day), was even higher: 104 subjects (99.0%) in the case group vs. 94 subjects (89.6%) in the control group (Fisher’s exact test, *p* = 0.005). However, according to Figure 3 and Appendix A, there was a statistically significant difference between the two groups when it comes to more specific categories of vitamin D intake (*p* < 0.001, Pearson chi-square test). In the case group, 95 subjects (90.5%) had vitamin D intake < 5 μg/day, while only 3 subjects (2.9%) had daily adequate intake according to the EFSA recommendations (ranging 17.4–30.2 μg/day), while in the control group, 11 subjects (10.5%) had an adequate vitamin D intake (ranging 15.2–43.3 μg/day), while 42 subjects (40.0%) and 41 subjects (39.0%) had vitamin D intake < 5 μg/day and 5–10 μg/day, respectively. None of the subjects had only through food a vitamin D intake higher than 100 μg/day (the upper tolerable limit).

In the whole sample, there were significant positive correlations between vitamin D intake and total energy, protein, fat, and calcium intakes (Spearman’s correlation coefficients, respectively, r_s_ = 0.550, *p* < 0.001, r_s_ = 0.693, *p* < 0.001, r_s_ = 0.619, *p* < 0.001, r_s_ = 0.624, *p* < 0.001), fish, eggs, dairy products, and added fats/oils intakes (r_s_ = 0.649, *p* < 0.001, r_s_ = 0.635, *p* < 0.001, r_s_ = 0.456, *p* < 0.001, r_s_ = 0.357, *p* < 0.001), serum vitamin D levels and physical activity levels (r_s_ = 0.483, *p* < 0.001, and r_s_ = 0.432, *p* < 0.001), and significant negative correlations between vitamin D intake and sugars/sweets intakes and smoking (r_s_ = −0.158, *p* = 0.022, and r_s_ = −0.248, *p* < 0.001, respectively). Vitamin D intake was not correlated with BMI and age.

The proportion of those who consumed vitamin D supplements at the time of examination for at least 3 months or longer was not significantly different: four subjects (3.8%) in the group with fractures vs. nine subjects (8.6%) in the control group (*p* = 0.152), nor was the average daily dose of vitamin D consumed through supplements (median dose of 50 μg/day in both groups). However, the number of those who consumed vitamin D-fortified food during the last year was much higher in the control group: 38 (36.2%) vs. only three (2.9%) in the case group, (*p* < 0.001). Additionally, three subjects in the case group consumed vitamin D supplements for the period of 1 month or less (after the fracture occurred and blood samples for vitamin D status were taken), with a daily dose of 150 μg/day, and therefore exceeded the recommended upper tolerable intake by both the EFSA and the IOM (100 μg/day).

In the case group, the median calcium dietary intake was significantly lower compared to the control group, 536.7 (420.4–688.9) mg/day vs. 945.0 (671.7–1192.2) mg/day (*p* < 0.001). At the same time, 99 subjects (94.3%) and 80 subjects (76.2%), in the case and control group, respectively, did not meet the recommended dietary intakes by the IOM recommendations (1200 mg/day) (*p* = 0.001), while according to the EFSA recommendations (950 mg/day), 73 subjects (94.3%) and 53 subjects (76.2%), in the case and control group, respectively, did not meet the recommended dietary intakes (0 < 0.001). Only two control subjects exceeded the upper tolerable limit of calcium intake by the EFSA (> 2500 mg/day, ranging from 2558.4–2693.5 mg/day) (Appendix A). None of the subjects reported the consumption of calcium supplements.

Calcium intake mostly correlated with dairy products intake, then with fish intake, less with grains, eggs, and added fats/oils intake (Spearman’s correlation coefficients r_s_ = 0.842, *p* < 0.001, r_s_ = 0.450, *p* < 0.001, r_s_ = 0.276, *p* < 0.001, r_s_ = 0.243, *p* < 0.001, r_s_ = 0.210, *p* = 0.002, respectively); with total energy, protein, fat, and carbohydrate intake (r_s_ = 0.653, *p* < 0.001, r_s_ = 0.754, *p* < 0.001, r_s_ = 0.555, *p* < 0.001, and r_s_ = 0.293, *p* < 0.001), and vitamin D intake and serum vitamin D levels (r_s_ = 0.624, *p* < 0.001, and r_s_ = 0.391, *p* < 0.001).

### 3.5. Serum Vitamin D Status

The serum vitamin D, defined as total 25(OH)D serum levels, in the control group were statistically significantly lower compared to the control group: median (IQR), respectively, 40.0 (23.0–50.0) nmol/L vs. 76.0 (57.0–91.0) nmol/L (Mann–Whitney test *p* < 0.001) (Figure 4 and Appendix A). The values were different even when the season was included as a covariate (*p* < 0.001, non-parametric ANCOVA). In fact, in total, the mean values of vitamin D were not different between samples taken from different sampling periods (seasons): for the late summer/early autumn period, samples median (IQR) values were for the total sample 46.0 (20.0–80.0) nmol/L, in the case group 19.0 (14.3–36.0) nmol/L, and in the control group 78.0 (58.0–95.0) nmol/L, while for the samples taken from the late autumn/early winter period median (IQR) values were for the total sample 50.0 (33.0–70.0) nmol/L, in the case group 34.0 (21.5–45.0) nmol/L, and in the control group 72.5 (56.0–84.0) nmol/L (Mann–Whitney test: the non-significant difference between seasons in the total sample and the control group, but in the group with fractures, the median levels were significantly lower in the late summer/early autumn samples, compared with the late autumn/early winter samples, *p* < 0.001, Figure 5). There was no difference in serum vitamin D levels between those subjects who consumed or did not consume vitamin D supplements at the time of examination, 68.5 (57.5–94.0) nmol/L vs. 77.0 (47.0–97.0) nmol/L, *p* = 0.243. The differences in vitamin D status remained significant even after the additional corrections for other covariates, including sex, age, BMI, smoking status, and physical activity level (non-parametric ANCOVA, *p* < 0.001).

A statistically significant difference between the case and control groups was also found by observing the distribution of subjects according to the scale of serum vitamin D reference values (*p* < 0001, Pearson chi-square test, Figure 6 and Appendix A), with all subjects in the case group having vitamin D values below the recommended values of 75 nmol/L (optimal value, according to the IOF, the American Geriatrics Society, and the Endocrine Society Task Force recommendations from 2011, which are now, in June 2024, abandoned) [46,49,50,51], while in the control group about half of the subjects (53 subjects, 50.5%) had values above 75 nmol/L (*p* < 0.001). The proportion of those who had suboptimal values of serum vitamin D, including levels of 50–75 nmol/L (vitamin D insufficiency) and 25–50 nmol/L (moderate vitamin D deficiency) among the case group was only seven subjects (6.7%) and 43 subjects (41.0%,), respectively, while 55 subjects (52.4% of the whole group) had a serious vitamin D deficiency (<25 nmol/L), associated with seriously disturbed bone health (osteomalacia, rickets, osteoporosis, and fractures) (Note: in the case group, eight subjects had values between 25–30 nmol/L, which is also associated with severe vitamin D deficiency, according to the IOM reference values; therefore, 63 subjects in total, i.e., 60.0% in the group with fractures, can be considered with a severe vitamin D deficiency). At the same time, corresponding proportions for vitamin D insufficiency, moderate vitamin D deficiency, and serious vitamin D deficiency in the control group were 38 subjects (36.2%), 12 subjects (11.4%), and only two subjects (1.9%) (Figure 6 and Appendix A). In the control group, three subjects (2.9%) had serum vitamin D values above 125 nmol/L (ranging from 127–133 nmol/L).

Vitamin D serum levels positively correlated with total energy, protein, fat, and fiber intake (respectively, r_s_ = 0.399, *p* < 0.001, r_s_ = 0.494, *p* < 0.001, r_s_ = 0.435, *p* < 0.001, and r_s_ = 0.177, *p* = 0.010), vitamin D and calcium intake (r_s_ = 0.483, *p* < 0.001, and r_s_ = 0.391, *p* < 0.001), with fish intake, less dairy products, eggs, and added fats/oils intake (r_s_ = 0.486, *p* < 0.001, r_s_ = 0.319, *p* < 0.001, r_s_ = 0.225, *p* = 0.001, and r_s_ = 0.292, *p* < 0.001), and negatively with sugars/sweets and alcoholic and non-alcoholic beverages intakes (r_s_ = −0.312, *p* < 0.001, and r_s_ = −0.173, *p* = 0.012). They also correlated positively with physical activity level (r_s_ = 0.527, *p* < 0.001), and negatively with smoking (r_s_ = −0.222, *p* < 0.001), but not with BMI.

### 3.6. Logistic Regression Models for Predicting the Risk of Fractures

To assess the importance of nutritional and non-nutritional factors as the risk factors for long bone fractures due to falls among elderly subjects, we performed several logistic regression analyses, applying different models, with the inclusion of predictors that showed significant differences between the two groups in our study, were significantly associated with odds for fractures in univariate models, or were reported in the literature to influence the risk for fractures, being careful to avoid multicollinearity issues (tested by VIFs) (Table 4). The univariate regression models for each of the selected candidate covariates were presented in Appendix A.

In univariate, non-adjusted Model 1, only vitamin D serum level was entered as a predictor variable. The odds ratio (OR) and its 95% confidence interval (CI) for serum vitamin D level were 0.884 (0.854–0.915), *p* < 0.001. Similarly, in multivariate Model 2, after adjusting for sex, age, BMI, smoking status, physical activity level, and season, serum vitamin D level was a significant protective factor, with OR (95%CI): 0.878 (0.841–0.916), *p* < 0.001, together with physical activity level (*p* = 0.006) (Table 4).

In the extended Model 3, also adjusted for sex, age, BMI, season, smoking status, and physical activity level, apart from vitamin D serum levels, additionally other nutritional variables that showed significant differences between the two groups were entered as predictor variables, including vitamin D dietary intake, vitamin D supplements intake, dietary calcium intake, dietary protein intake, dietary fat intake, and dietary fiber intake (dietary energy intake was not entered due to multicollinearity issues). The stepwise regression model again identified the serum vitamin D status as the most significant protective factor, OR (95%CI): 0.864 (0.822–0.909), *p* < 0.001, but also dietary intakes of protein and vitamin D were significant protective factors, with OR (95%CI): 0.939 (0.907–0.973), *p* < 0.001 and 0.895 (0.796–1.006), *p* = 0.064, respectively (Table 4).

Since there was a high co-linearity between fat and protein intake (Spearman’s correlation coefficient r_s_ = 0.766, *p* < 0.001), and stepwise models do not allow co-linearity, we excluded protein intake as a candidate covariate from the stepwise regression model (Model 4). In the case protein intake was excluded from the model, fat intake was significantly associated with decreased odds for fractures, OR (95%CI): 0.966 (0.941–0.992), *p* = 0.012, together with fiber intake, OR (95%CI): 0.841 (0.739–0.958), *p* = 0.009, and vitamin D intake, OR (95%CI): 0.900 (0.800–1.012), *p* = 0.079. Serum vitamin D level was again the most significant protective factor, OR (95%CI): 0.857 (0.808–0.909), *p* < 0.001 (Table 4).

In the extended Model 5, again adjusted for sex, age, BMI, season, smoking status, and physical activity level, instead of the macronutrient dietary intakes, caloric intakes from specific food groups were entered, including dietary milk and dairy product intakes, egg and egg product intakes, fish and seafood intakes, dietary edible fats and oils intakes, and sweets and added sugars intakes. The stepwise regression model again identified the serum vitamin D status as the most significant protective factor, OR (95%CI): 0.794 (0.713–0.885), *p* < 0.001, followed by fish intake, OR (95%CI): 0.972 (0.958–0.987), *p* < 0.001, egg intake, OR (95%CI): 0.982 (0.969–0.996), *p* = 0.011, and milk and dairy products intake, OR (95%CI): 0.992 (0.985–0.999), *p* = 0.034, while intake of sweets and added sugars was a significant risk factor, OR (95%CI): 1.013 (1.002–1.025), *p* = 0.022 (Table 4).

In the last three models (Model 3, Model 4, and Model 5), the late autumn/early winter season was an additional risk factor (compared to the late summer/early autumn season), while physical activity level was a protective factor in Model 4 and Model 5 (Table 4). It is important to emphasize that the models explained 75.5% (Model 1) to 91.9% (Model 4) of the variability in the risk for fractures among elderly people in this study (Table 4).

### 3.7. ROC Curve Analysis for Serum Vitamin D Cutoff Values for the Increased Risk for Fractures

Since serum vitamin D was a significant predictor in our univariate regression model, and the serum vitamin D data did not follow the normal distribution in both groups, we performed the nonparametric ROC curve analysis for serum vitamin D cutoff values, under which the risk for fractures will be significantly increased. The analysis indicated the serum vitamin D value of 50.5 nmol/L as a cutoff point (using both Youden’s J statistics and maximum multiplication of sensitivity and specificity approaches), with a sensitivity of 93.3%, specificity of 86.7%, and accuracy of 90.0% (area under the curve, AUC (95%CI): 0.950 (0.924–0.977), standard error (SE): 0.014, *p* < 0.001) (Appendix A).

## 4. Discussion

The present study examined the dietary characteristics and vitamin D status among elderly people with and without fractures in the Vojvodina region of Serbia and found significant differences between the two groups in terms of vitamin D status, macronutrient and micronutrient intakes (particularly of total energy, protein, fat, fiber, calcium, and vitamin D), as well as intakes of certain food groups (fish, eggs, dairy products, added fats/oils, and sweets/added sugars), smoking habit, and physical activity. The serum vitamin D level was the most important protective factor for fractures, with values above 50.5 nmol/L being associated with a significantly decreased risk for fractures. Other significant nutritional protective factors were the intake of proteins, fats, fibers, and vitamin D as well as the intake of fish, eggs, and dairy products. In contrast, the intake of sweets/added sugars was a risk factor.

In this study, we used sex-matched cases and controls, with about three-quarters of women included, which is in accordance with the higher prevalence of osteoporosis and long-bone fractures due to falls among older females, compared to older males (ratio 3:1) [1,2]. Elderly men, in general, have initially higher bone and muscle mass, are physically more active, are more hormonally protected (without the abrupt effect of menopause and loss of estrogens), and have a shorter life expectancy, compared to elderly women, and thus have a lower incidence of osteoporosis and fractures [56,57,58,59,60]. The groups of cases and controls were comparable in age and BMI: in terms of age, the subjects were on average only 2 years older in the case group (73 vs. 71 years), and in terms of BMI, only slightly heavier in the control group (26 vs. 27 kg/m^2^), and in both groups about 83% belonged to normal-weight and overweight. However, the groups were significantly different in terms of total daily energy intake, with about 450 calories lower intake in the case group (~1300 kcal vs. ~1750 kcal), which was most probably compensated by significantly lower physical activity in this group, with 77% being intermediate active or moderately/highly inactive, while in the control group, even 78% being moderately/highly active. In accordance, energy intake was only weekly associated with BMI. Even though some researchers found that a moderately higher BMI can be associated with a lower risk of fractures [17,61,62,63], our study did not confirm nor refute such a statement, since our subjects were quite similar in terms of BMI, and BMI was a significant predictor only in the univariate model. Chronic caloric restriction not only decreases bone mineral density (BMD) but also increases bone marrow adiposity, which is also associated with a higher risk for fractures [62,63,64,65,66].

In our study, the differences in total caloric intake between the two groups were mostly based on the reduction of protein intake and fat intake. Proteins are necessary for building bone matrix and for muscle mass and strength since they increase insulin-like growth factor-1 (IGF-1), a polypeptide hormone that regulates bone and muscle formation, and calcium and phosphorus metabolism [18,19,67]. Additionally, they increase the intestinal absorption of calcium [68]. However, they can increase kidney calcium excretion, if consumed in very high amounts, but the effect on bone is mostly not detrimental, since the proportion of the calcium deriving from bones in the urine is reduced in such conditions [67,68]. Muscle mass and strength not only prevent falling [69,70] but also directly influence, mechanically and metabolically (by secreting myokines), the formation of bones and thus prevent osteoporosis and fractures [63,71,72,73]. However, some meta-analyses did not show a significant association between total protein intake and risk for osteoporosis or bone fractures in the general population [66,67]. Nevertheless, the nutrition of older people is often characterized by low protein intake, low level of protein digestion and absorption, lower incorporation in the muscle and bone (“anabolic resistance”), and increased loss of nitrogen due to hypercatabolism and physical inactivity [55,74]. For those reasons, in order to achieve a nitrogen balance and avoid a progressive loss of lean mass, the increased protein intake was proposed by the European Society for Clinical Nutrition and Metabolism (ESPEN) of at least 1.0 to 1.2 g protein/kg body weight/day for healthy older people, or 1.2 to 1.5 g protein/kg body weight/day for older people who are malnourished or at risk of malnutrition because of acute or chronic illnesses, with a further increase in case of more serious illnesses, together with increased physical activity or exercise [74]. In our study, in the case group, ~77% had a protein intake lower than 0.8 g protein/kg body weight/day, and ~84% had a protein intake lower than 1.0 g protein/kg body weight/day, indicating insufficient protein intakes [54,55], while in the control group, ~71% had a protein intake higher than 0.8 g protein/kg body weight/day, and ~49% had a protein intake higher than 1.0 g protein/kg body weight/day.

Fats in nutrition are important for intestinal absorption of liposoluble vitamins, including vitamins D and K, important for bone health. On the other hand, depending on the type of fat, they can impair calcium intestinal absorption, by making insoluble soaps (mostly saturated fats) [75], or increase calcium absorption by stimulating its transport system (e.g., omega-3 and monounsaturated fats, but also saturated fats) [76,77]. Omega-3 fatty acids are important for cognitive performance and healthy and stable joints, and probably can strengthen the bone matrix, and prevent osteoporosis and bone marrow adiposity (which is mostly supported by some epidemiological and animal studies, but still not confirmed by interventional trials) [67,78,79,80,81,82,83,84,85], while intakes of saturated and trans-fatty acids were shown to be associated with an increased risk for fractures and osteoporosis in some epidemiological studies, but not in all, and even decreased risk was associated with saturated fats [67,77,86,87,88,89,90]. Similarly, the intake of monounsaturated fats was associated with increased calcium absorption and BMD in some but not all studies, and some indicated an even increased risk for fractures associated with animal monounsaturated fats [67,87,88,89], while the intake of omega-6 polyunsaturated fats was more inconsistently associated with calcium absorption, BMD, and bone marrow adiposity, being positively, neutrally, or negatively associated in epidemiological and animal studies [67,80,81,82,89]. Saturated, monounsaturated, and omega-3 and omega-6 polyunsaturated fats, either directly or through their active metabolites-eicosanoids, influence bone structure mostly by conveying their pro- or anti-inflammatory properties, cytokine production, oxidative stress, affecting lipoproteins, adipokines, insulin resistance and IGF-1 levels, osteocalcin, bone collagen synthesis, nuclear factor kappa-β (NF kappa-β), receptor activator of NF kappa-β (RANK) and its ligand (RANKL), and osteoprotegerin (OPG), the latter three involved in osteoclast function [79,85,91]. As said, they can both increase and decrease calcium absorption [75,77,92]. In general, too high intake of fat was associated with decreased BMD, increased bone marrow adiposity, pro-inflammatory environment, and therefore risk for fractures, while moderately high fat intake was associated with better BMD [66,76,93,94]. In our study, most of the fat in the diet in both groups was coming from added oils and fats (in the first-place sunflower oil, making on average 20–22% of total energy coming from fat), then meat and dairy products, but we could not distinguish specific fatty acids contribution, due to incompleteness of the used food composition database. In our regression model with protein intake included, fat intake was not a significant protective factor for fractures, due to high co-linearity with protein (r_s_ = 0.766), vitamin D (r_s_ = 0.619), and calcium (r_s_ = 0.555) intake (*p* < 0.001 for all). However, when protein intake was excluded from the model, fat intake was significantly associated with decreased odds of fractures. Nevertheless, a clear distinction of the effect of fat cannot be made, if it was a simple confounding factor or a real preventive factor.

Similarly, in our study, also dietary fiber intake was positively associated with a decreased risk for fractures, but only when the protein intake was not taken into account. Dietary fiber intake was positively associated with a decreased risk for osteoporosis also in the literature [20,95,96], although not in all studies [97,98]. Some studies indicate the inverse U-shaped association of BMD with dietary fiber intake [98]. Particularly, the ratio of dietary fiber to carbohydrates was associated with a decreased risk for osteoporosis [20] and can prevent bone marrow adiposity and fracture risk [66,99]. Fibers are prebiotics, e.g., inulin or precursors for short-chain fatty acids (e.g., propionate, butyrate, acetate), while short-chain fatty acids, gut microbiota, and probiotics are also associated with calcium absorption, inflammation, osteoblasts function/bone formation, osteoclasts function/bone resorption, and thus BMD [95,96,100,101,102,103]. On the other hand, high phytate and oxalate content associated with fiber intake can decrease calcium absorption [104,105]. In our study, there was a very high and significant positive association of dietary fiber intake with vegetable and fruit intake (but no association with cereals intake, since the majority of the subjects consumed only refined grain products), and a small but still significant positive association with meat, nuts/seeds, and fats/oils intakes, as well as with total protein and fat intake and vitamin D status, and a small but significant negative association with intake of sugars, sweets, and beverages. For those reasons, the intake of fiber can be confounded by other important micro- and macronutrients. Some vegetables and fruits are sources not only of dietary fiber, but also of vitamin K, carotenoids, folate, magnesium, and potassium, which are all important for bone health [18,19], so the effect of dietary fiber alone cannot be clearly distinguished.

In our study, we did not show the effect of carbohydrates, which is in accordance with other studies [20,67]. Calcium intake is significantly associated with decreased risk for osteoporosis and bone fractures in the majority of studies, but not all [18,19,106,107,108,109,110]. However, in our study, it was a significant predictor only in the univariate model, but not in the multivariate model, most probably due to high multicollinearity with vitamin D status, and vitamin D, protein, fat, and fiber intake. However, the intakes of dairy products, which were highly associated with calcium intake, were significantly associated with decreased odds for fractures, which is in accordance with other literature data [18,19,111].

Vitamin D is a liposoluble vitamin that keeps the blood calcium and phosphorus levels at the optimum, and this way indirectly promotes the deposition of calcium in the bones. The main target tissues of its active form, 1,25(OH)_2_D, related to skeletal health are the intestine, the kidneys, the parathyroid glands, and the bones, where it binds to its nuclear and membrane vitamin D receptors (VDRs) and other membrane receptors (e.g., G-protein coupled receptors) to stimulate calcium and phosphorus absorption (in the intestine), the tubular calcium reabsorption, together with PTH (in the kidneys), to decrease parathyroid hormone (PTH) secretion (in the parathyroid glands), and increase bone resorption by osteoclasts, together with PTH (in the bones) [47]. In the case of vitamin D deficiency, PTH increases, which leads to osteoporosis [112]. Additionally, vitamin D is important for muscle strength and function and has many other roles (metabolic, cardiovascular, immune, anti-inflammatory, mental, reproductive, etc.) [47,113]. Its stores in the body are short-lived (only 6 to 8 weeks, serum half-life: approximately 8 to 13 days) and are found mostly in adipose tissue, liver, and muscle [114]. The synthesis of vitamin D3 in the skin depends on the availability of its precursor 7-DHC, the amount of UV-B irradiation reaching the dermis, and heat (body temperature), and is generally decreased with advancing age [45,47]. During the summer months, the main source of vitamin D could be the endogenous synthesis of vitamin D3 in the skin, but it should not be a priori assumed (since it depends on the latitude, altitude, UV index, cloudiness, lifestyle habits, religious norms and exposure of the skin to sun, skin color, age, etc.), and dietary intake of vitamin D is essential in the case when endogenous synthesis is scarce, e.g., due to insufficient UV-B exposure or age [45,47]. Significant proportions of the European population, particularly those residing above 40° N latitude, depend on the stored body reserves and dietary intakes of vitamin D to maintain adequate levels throughout the year [115]. With many parts of Europe enduring 4–6 months of winter with insufficient UV-B exposure for synthesizing vitamin D3 through the skin, reliance on dietary sources becomes imperative to fulfill the population’s needs. Elderly people particularly face the risk of inadequate vitamin D levels due to reduced mobility and outdoor activities, increased clothing, and generally less time spent exposed to the sun [115]. The novel IOM and Endocrine Society recommendations suggest for older adults an intake of at least 20 μg/day [45,46], while the EFSA recommendations still rely on 15 μg/day [47]. The Endocrine Society, in their newest guidelines, also recommends a routine vitamin D supplementation for adults ≥70 years old [46]. In our study, the average intake of vitamin D in both groups was far below the recommended levels by both the EFSA and the IOM/Endocrine Society [45,46,47], 1.4 (0.9–2.7) vs. 5.8 (3.3–8.6) μg/day, in the case and control group, respectively, although the percentage of those who satisfied the EFSA and the IOM/Endocrine Society [45,46,47] criteria was significantly higher in the control group (14.3% and 10.4%, respectively), compared with only 2.9% and 1.0%, in the case group. However, the serum vitamin D levels satisfied the standing EFSA/IOM recommendations (50 nmol/L) in 86.7% of the controls, with average values far above this threshold. The adequate serum vitamin D levels in the control group, despite the very low vitamin D intake, can be explained by the endogenous synthesis, probably due to higher physical activity and exposure to the sun, but also by the fact that only two 24-h dietary recalls were used, which can underestimate the vitamin D intake at the individual level, since vitamin D food sources are very limited, and day-to-day variability in the intake is very high. In contrast, only 6.7% of the cases satisfied the current EFSA/IOM targets for serum vitamin D levels, which is in accordance with their lower vitamin D intakes, decreased physical activity, and probably lower exposure to the sun. For comparison, it is estimated that in Europe, 27.2% to 61.4% had serum vitamin D levels < 50 nmol/L, while mean intake of vitamin D varied from 4 to 14 μg/day in Northern Europe, from 1.5 to 5 μg/day in Western Europe, and below 1 μg/day to about 3 μg/day in Southern Europe [116]. Vitamin D intake is highest in the Nordic countries and the lowest in southern Europe due to a diet traditionally more based on fish, a raised awareness of problems of vitamin D deficiency, and thus an established practice of consumption of vitamin D supplementation and fortification of a range of foods, compared to central and southern European countries [117]. Subjects in our study had a typical European continental diet, mainly based on refined grains, edible fats, non-fortified dairy products, and red meat, with low consumption of fish and offal (which are good sources of vitamin D), and very moderate consumption of eggs.

Our study identified that serum vitamin D levels below 50.5 nmol/L were significantly associated with a higher risk for fractures (with high accuracy, sensitivity, and specificity), and similar values were often identified in the literature [47,116]. Vitamin D not only influences calcium absorption/reabsorption and BMD (thus decreasing the risk of rickets, osteomalacia, osteoporosis, and bone fractures, predominantly in the forearm, vertebrae, and hip), but also influences muscle function and strength, physical performance, mental function, and prevents falls [47,69]. Other studies also examined the critical values of serum vitamin D among elderly subjects related to low BMD and risk for osteoporosis, the risk for fractures, the risk for falls, impaired muscle strength/function/physical performance, and decreased calcium absorption [47]. The results were quite different, e.g., critical serum vitamin D values varied from 30 to 80 nmol/L (for BMD), 18 to 80 nmol/L (for fractures), 25 to 75 nmol/L (for muscle strength/physical performance), 23 to 82 nmol/L (for falls), 20 nmol/L (for osteomalacia), and 10 nmol/L (for calcium absorption), but in the majority of them, the cutoff for the increased risk for osteoporosis, fractures, falls, and muscle weakness/dysfunction was around 50 nmol/L. In line with such findings, the EFSA panel (2016) set this cutoff value for normal musculoskeletal health in adults, including the elderly [47], and the European Society of Clinical and Economical Aspects of Osteoporosis, Osteoarthritis and Musculoskeletal Diseases (ESCEO) adopted it [116]. This cutoff is closely in accordance with the findings of our study. It is worth noting that the USA Endocrine Society from June 2024 [46,49] also refers to the IOM cutoffs of 50 nmol/L from 2011 and no longer endorses 75 nmol/L as the targeted value of serum vitamin D (suggested in their previous guidelines from 2011 [49], also in the guidelines for elderly people by the IOF from 2010 [50] and the American Geriatrics Society from 2014 [51]). Of interest, one study has found that the risk for fractures had a U-shaped association with serum vitamin D levels, with values both < 36 nmol/L and > 72 nmol/L being associated with the increased risk for fractures among older men [118], while many of the studies did not show any beneficial effects of vitamin D supplementation on the decreased risk for fractures [113,116]. Furthermore, different meta-analyses of the randomized controlled trials (RCTs) with vitamin D supplementation on the fracture risk have revealed contradictory results, with some of them showing a non-significant effect, while in contrast, some showing a beneficial effect, particularly with doses ≥ 20 µg/day and when combined with calcium supplementation (minimum of 500 mg/day) [113,119,120]. The reason for the contradictory results was the selection of the studies included the following: the included studies used different doses (10 µg/day to 12,500 µg/annually), different dosage schemes (daily, weekly, monthly, 4-monthly, or annually), administration routes (orally vs. intramuscularly), different durations of treatment (from 1 year to 7 years), the addition of calcium or not, the addition of estrogens or not, different sex, age, settlement type (institutionalized or community-based free-living), and other characteristics of the subjects included (e.g., BMI: in obese subjects, higher doses are needed due to volumetric dilution), different compliance rates, different starting level of vitamin D in serum, and different vitamin D levels in the serum achieved [113,119,120]. In many of the RCTs, the serum vitamin D levels and BMD achieved were not sufficient to decrease the fracture risk, probably due to insufficient duration, doses used, compliance rate, or calcium intake. Regarding the effects of vitamin D supplementation on the BMD, they were the most prominent at the femoral neck [119]. Additionally, the U-shaped association between vitamin D levels and risk for fractures was shown [116,118]. For those reasons, after a careful consideration of meta-analyses of RCTs, for the prevention of fractures, the ESCEO in 2022 suggests a beneficial effect of vitamin D supplementation (in doses 20–25 µg/day, together with calcium supplementation), only for adults with 25(OH)D levels < 50 nmol/L, to bring them into a 25(OH)D range of 50–100 nmol/L, while no beneficial effect of supplementation for adults with 25(OH)D levels already in the range of 50–100 nmol/L, and implies that vitamin D supplementation may even increase the risk of fractures if resulting in 25(OH)D levels > 100 nmol/L [116]. In contrast, the USA Endocrine Society in 2024 advises a routine supplementation with 20 µg/day for all adults ≥ 70 years old, without measuring the serum vitamin D levels [46]. For our population, according to our study results, measuring the serum vitamin D levels before supplementation probably would be a better approach, with doses of 20–25 µg/day.

The differences in vitamin D intakes were in accordance with the observed differences in the intake of food groups that are the major sources of vitamin D, including fish, eggs, dairy products, and higher consumption of vitamin D-fortified food. In our study, the intakes of fish, eggs, and dairy products were significantly associated with decreased odds for fractures, which is in accordance with most studies [78,111,121,122,123,124,125,126,127,128,129,130,131,132], although some other studies did not find similar associations or the effect was modest, particularly for some dairy products [18,111,127,128,133,134,135,136,137]. Fish is a good source of vitamins D and A, calcium (e.g., sardines), zinc, copper, magnesium, selenium, iodine, omega-3 fats, and proteins, and all of them are important for bone health [18,123]. In our study, fish intake was also positively associated with those nutrients, as well as with serum vitamin D levels (*p* < 0.001 for all). Eggs are also a good source of vitamin D, carotenoids, and proteins, including egg white protein ovotransferrin, which can prevent bone resorption by osteoclasts and stimulate osteoblast activity through regulation of the OPG/RANKL ratio [138,139,140]. In our study, egg intake was positively associated with vitamin D, calcium, protein, and fat intake, as well as with serum vitamin D levels (*p* < 0.001 for all). Dairy products are not only the major source of calcium in the diet, but also a good source of phosphorus, valuable proteins, vitamin D (if fortified), vitamin K2 (e.g., cheese), or probiotics (e.g., yogurt). In our study, also dairy product intake was positively associated with vitamin D, calcium, protein, and fat intake, as well as with serum vitamin D levels (*p* < 0.001 for all). Edible fats and oils intakes were also associated with decreased odds for fractures in the univariate model, but not in the multivariate model, and the effect can be confounded by the moderate co-linearity with the intake of all other micro/macronutrients and food groups shown here as protective factors (protein, fat, fiber, calcium, vitamin D, milk, egg, and fish) and vitamin D status (*p* < 0.01 for all). On the other hand, the intake of sweets/sugars and alcoholic and non-alcoholic beverages was associated with increased odds for fractures (the latter only in the univariate model), which can be explained by their negative associations with vitamin D status, other micro- and macronutrients associated with bone health and lower risk for fractures, physical activity, and positive association with smoking (*p* < 0.05 for all). Additionally, although higher simple sugar intakes were associated with both detrimental and beneficial effects on bone health, with sucrose having mostly detrimental effects [66,76], the low ratio of dietary fiber to carbohydrates (typical for sweets and sugared beverages) and ultra-processed food consumption were associated with osteoporosis and risk for fractures [20,66,99]. Ultra-processed food is also high in phosphorus (e.g., cola), and an inappropriate ratio of calcium to phosphorus can increase PTH secretion and decrease calcium absorption and incorporation in bones [141,142,143]. Heavy alcohol consumption, with alcohol intake of ≥27 g (i.e., >two standard drinks) per day, has been associated with lower BMD and increased risk for fractures [61]. However, some data indicate that a lower to moderate alcohol consumption (i.e., ≤one standard drink) may be associated with increased BMD and decreased risk for fractures [144].

Smoking can be associated with other nutritional and non-nutritional modifiable factors related to risk of falls, osteoporosis, and fractures (also in our study, correlation with all other significant factors, *p* < 0.01 for all), but itself can increase the risk for osteoporosis and fractures, by affecting the OPG/RANKL-RANK pathway and osteoblast and osteoclast proliferation, differentiation and apoptosis, bone vascularization, by inducing oxidative stress and pro-inflammatory cytokine releases, by defective collagen synthesis, by decreasing estrogens and increasing adrenal hormones (cortisol), by modulating PTH, calcitonin, and decreasing vitamin D levels and hydroxylation, leading to decreased calcium absorption, reabsorption, and disturbed calcium-phosphate balance, and by affecting appetite, body weight, and intestinal microbiota composition [61,132,145,146,147]. In our study, smoking was negatively correlated with energy intake, intake of proteins, fats, fibers, calcium, vitamin D, milk products, eggs, fish, physical activity, and serum vitamin D levels, and positively with intake of added sugars (*p* < 0.001–0.007), and because of this high co-linearity with other significant lifestyle factors, smoking was not a significant predictor in multivariate regression models (only in the univariate one), but this does not exclude its possible influence.

Exercise and physical activity were shown to be positively associated with BMD (since physical load, mechanical and gravitational effect of weight and muscle mass, and myokines increase bone matrix formation and mineralization) and negatively associated with bone marrow adiposity, and can influence gut microbiota [62,132,148,149]. In accordance, in our study, the level of physical activity was a significant independent predictor in multivariate models. Nevertheless, physical activity was also positively correlated with total energy intake, intake of proteins, fats, fibers, calcium, vitamin D, milk products, eggs, fish, added fats and oils, and serum vitamin D levels, and negatively with the intake of added sugars and smoking (*p* < 0.001–0.035), indicating also an indirect effect. Particularly, serum vitamin D levels were significantly correlated with physical activity (r_s_ = 0.527, *p* < 0.001). Physical activity can be associated with increased sun exposure (in the case of outdoor activities) and increased lipolysis and release of vitamin D from adipose tissue stores into circulation [150,151].

The late autumn/early winter season was an important risk factor, which is in accordance with the data about the incidence of falls, fractures, and deaths because of falls among elderly people during one year through different months [152,153]. However, there was a small selection bias, because slightly more cases were recruited in September, while the selection of controls went a bit after. For that reason, we had to enter the season as a covariate when examining the effect of vitamin D, even though there was not a significant difference in the season’s frequency between cases and controls. In the Vojvodina region (between 41°53′ and 46°11′ N), “the vitamin D winter” period is considered to be from November through February [47]. Nevertheless, it is an interesting and unexpected finding that among cases, vitamin D serum levels were lower in the late summer/early autumn period (September) than in the late autumn/early winter period (November through January). One of the explanations can be that older people synthesize less endogenous vitamin D when exposed to the sun, and many of them tend to avoid the sun and physical activity during summer, which places them at higher risk for vitamin D deficiency and fractures. However, we did not collect data on sun exposure to compare the two groups. Higher vitamin D levels in the winter in the case group can be explained by a higher vitamin D intake in the late autumn/early winter period in this group, compared with the late summer/early winter period.

### Study Strengths and Limitations

The strength of our study is that it examined many factors related to the risk of fractures and allowed one comprehensive analysis while adjusting for multiple covariates. However, there are several study limitations. Case-control studies can only establish associations/correlations, but not also causation, they can introduce a recall bias, and there is a need for an appropriate control group, which can introduce a selection bias. In our study, we matched cases and controls only for sex and geographical region of residence, but not for other key characteristics (additionally, all subjects were community-based free-living, non-institutionalized, and groups were similar according to the urban vs. rural type of settlement). However, our groups also were not very different in terms of age, BMI, other demographic characteristics, or season of examination. Furthermore, we also performed all analyses with adjustments for sex, age, BMI, physical activity, smoking, and season (or only with adjustments for sex, age, and BMI-) to eliminate their possible confounding (by applying ANCOVA or partial correlations), and the results were not different. Another study limitation is the relatively small sample size, which did not allow sufficient power for logistic regressions in conditions when the obtained ORs were in the range of 0.6–1.7, particularly in the case when multiple, possibly inter-correlated variables were tested in the model [154]. An additional study limitation is that CMIA was used to determine vitamin D status, while the golden standard is liquid chromatography linked to tandem mass-spectrometry detectors (LC-MS/MS) and high-performance liquid chromatography linked to an ultra-violet detector (HPLC-UV) [45]. However, CMIA is a standard technique used to measure vitamin D in clinical settings, due to its simplicity, time efficiency, and economic costs; it has an acceptable sensitivity, specificity, accuracy, and precision (within-run and between-run repeatability); and a high correlation with the HPLC-UV method was shown [32]. Another study limitation is that only two 24-h dietary recalls were used to assess the nutritional intakes, which do not have to represent the habitual diet of the subject, due to day-to-day variation. This is particularly related to the assessment of vitamin D intake since this nutrient is not found in many food groups typical for Serbian cuisine, and maybe the vitamin D intake at the level of population can be assessed relatively accurately by only two recalls (which is valid in case of examining the difference between two groups), but at the individual level (in case of estimating the risk in logistic regression), it is not precise enough, and more recalls are needed (e.g., four), or a validated food frequency questionnaire [155]. An additional problem of 24-h dietary recalls is that they rely on memory (ability to remember food intake) and the ability to evaluate portion sizes, and among older subjects, both memory and cognitive performances are often impaired, and therefore data should be interpreted with caution [156]. However, we estimated that the possible errors would be randomly distributed between the two groups. Furthermore, there is a lack of national dietary recommendations to compare our data, and we used foreign recommendations for older males and females, which may not be quite applicable to our population. Another study limitation is that we could not assess some other nutrients related to bone metabolism, e.g., vitamin K, magnesium, potassium, folate, carotenoids, and different fatty acids, due to methodological issues and incomplete data in the used food composition table. As previously said, we also did not collect data on sun exposure, which would also be very informative. Additionally, we did not measure the BMD to examine the level of osteoporosis among our case and control subjects, due to the limited capacities of our public health system, lack of available instruments, and the heavy condition of subjects with fractures to be transported to another clinic for measurement. Unfortunately, in Serbia, there is no regular screening of BMD among older people, which could be recommended. Because we did not measure BMD, we cannot relate our conclusions firmly to osteoporosis. However, our study results indicate such an association and we made a careful selection of the subjects with strict inclusion/exclusion criteria, by which were eliminated other possible important causes of fractures (e.g., bone metastases, kidney diseases, endocrine diseases, use of certain medications, etc.).

## 5. Conclusions

Vitamin D status, dietary patterns, and dietary intakes of certain macronutrients and micronutrients can be associated with a lower risk of fractures among elderly subjects in the Vojvodina region of Serbia. The study showed a significant association between serum vitamin D levels, dietary intakes of vitamin D, calcium, protein, fat, and fiber, and consumption of fish, eggs, and dairy products, with a lower risk for fractures, and consumption of sweets with an increased risk. Vitamin D status was the most important determinant of the risk, with 25(OH)D values above 50.5 nmol/L being associated with significantly decreased risk. However, the dietary vitamin D intake in the examined elderly population was generally low. Having in mind a high socioeconomic burden and increased mortality risk connected with long bone fractures among the elderly population, changes in dietary habits, with the enhancement of intake of fish, eggs, dairy products, offal, and vitamin D-enriched foods, as well as regular sun exposure of the hands, feet, and face of at least 30 min per day (during the spring and summer months) and preventive vitamin D supplementation (during autumn and winter, particularly in those with vitamin D serum levels below 50 nmol/L), together with moderately increased physical activity and smoking cessation, can be important strategies in the prevention of fractures among elderly subjects in this region.

## Figures and Tables

**Figure 1 nutrients-16-02702-f001:**
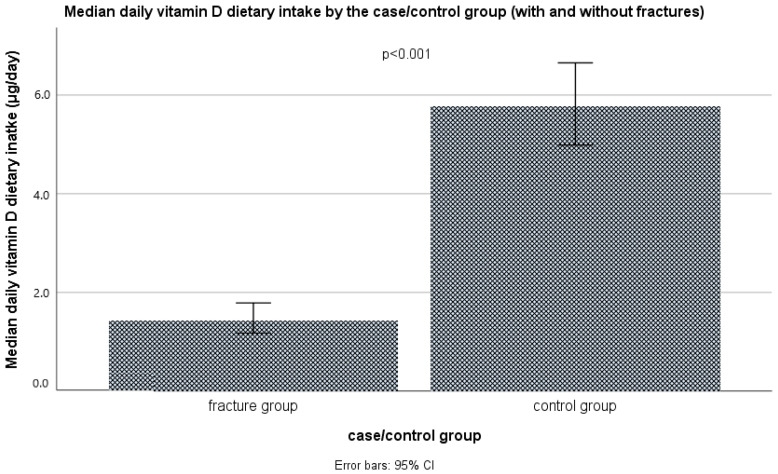
The average intake of vitamin D (μg/day) through diet in the case and control group, Mann–Whitney test.

**Figure 2 nutrients-16-02702-f002:**
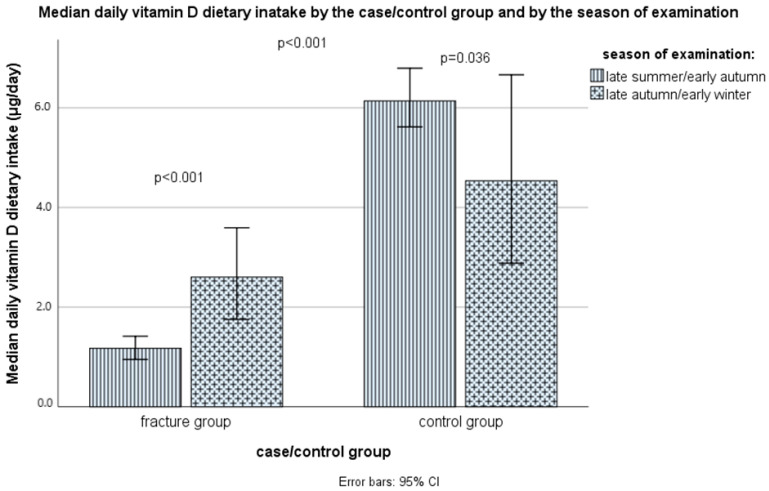
Average (median) daily vitamin D intakes (μg/day) through diet in the case and control group, separated by season of sampling, Mann–Whitney test.

**Figure 3 nutrients-16-02702-f003:**
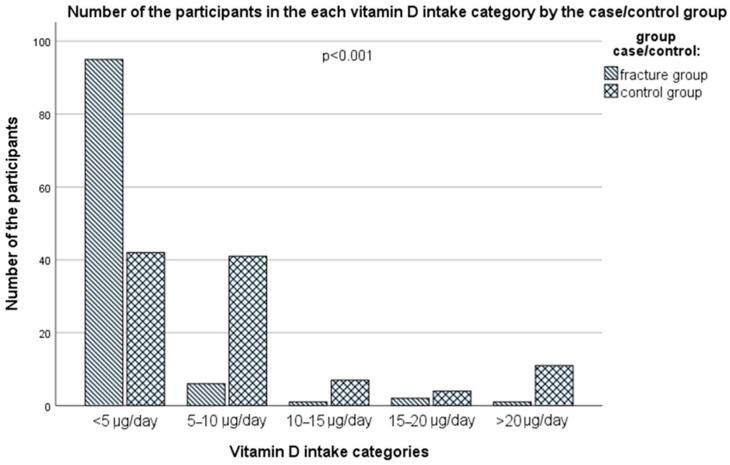
Categories of intake of vitamin D (μg/day) relative to recommended values, Pearson chi-square test.

**Figure 4 nutrients-16-02702-f004:**
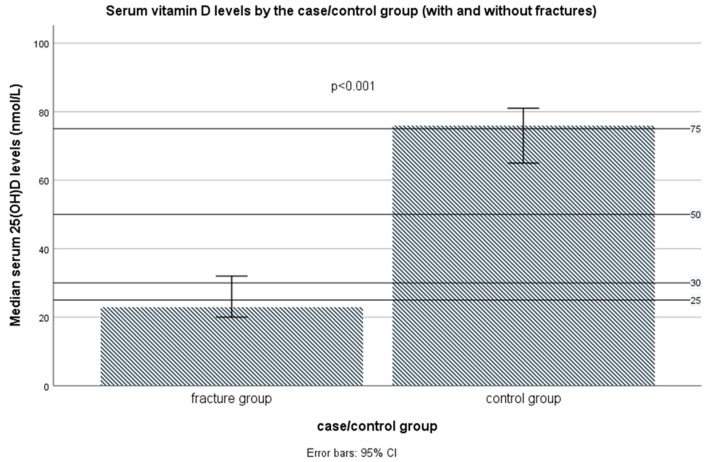
Average (median) serum 25(OH) vitamin D levels (nmol/L) in the case and control group, Mann–Whitney test.

**Figure 5 nutrients-16-02702-f005:**
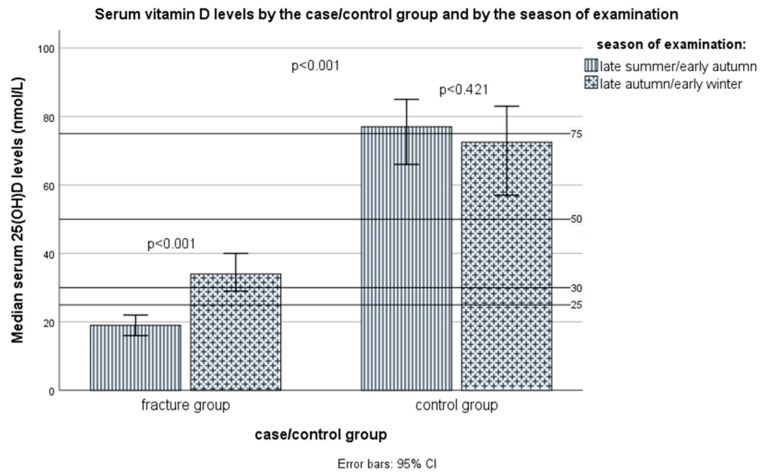
Average (median) serum 25(OH) vitamin D levels (nmol/L) in the case and control group, separated by season of sampling, Mann–Whitney test.

**Figure 6 nutrients-16-02702-f006:**
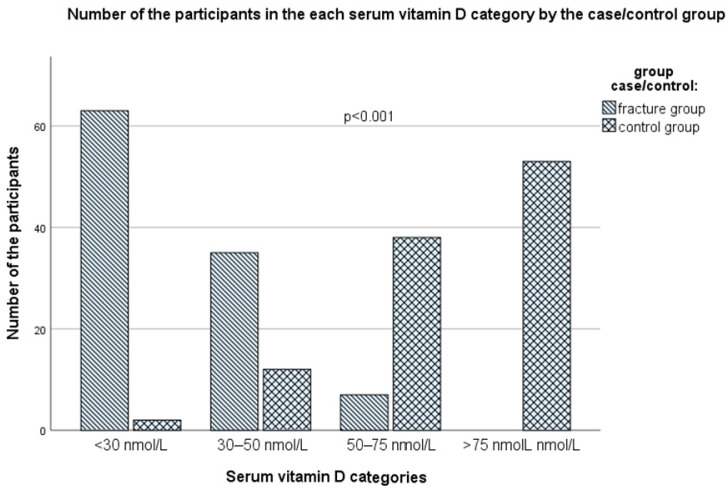
The categories of the subjects based on serum vitamin D status in the case and control group, the Pearson chi-square test.

**Table 1 nutrients-16-02702-t001:** Comparison of socioeconomic, lifestyle, anthropometric, and medical data between elderly subjects (>65 years) with and without fractures in the Vojvodina region, Serbia.

	With Fractures (*n* = 105)		Controls (*n* = 105)		
	Median/*n*	(IQR)/(%)	Median/*n*	(IQR)/(%)	*p*
Sex:					1.000
• Men *n* (%)	15	(23.8%)	15	(23.8%)	
• Women *n* (%)	80	(76.2%)	80	(76.2%)	
Age (years)	73.0	(69.0–78.0)	71.0	(69.0–75.0)	**0.044**
Education:					0.440
• Primary education (ISCED 1) or lower *n* (%)	15	(14.3%)	17	(16.2%)	
• Lower secondary education (ISCED 2) *n* (%)	27	(25.7%)	16	(15.2%)	
• Upper secondary education (ISCED 3) *n* (%)	35	(33.3%)	36	(34.3%)	
• Post-secondary but non-tertiary education (ISCED 4) *n* (%)	13	(12.4%)	17	(16.2%)	
• First stage of tertiary education (ISCED 5) *n* (%)	13	(12.4%)	14	(13.3%)	
• Second stage of tertiary education (ISCED 6) *n* (%)	2	(1.9%)	5	(4.8%)	
Settlement type:					0.774
• Rural	37	(35.2%)	39	(37.1%)	
• Urban	68	(64.8%)	66	(62.9%)	
Marital status:					0.324
• Married	77	(73.3%)	83	(79.0%)	
• Widowed, single, divorced	29	(27.7%)	22	(21.0%)	
Household person number	2.0	(2.0–4.0)	2.0	(2.0–4.0)	0.766
Smoking status:					**<0.001**
• Never smoker	12	(11.4%)	43	(41.0%)	
• Former smoker	45	(42.9%)	38	(36.2%)	
• Current smoker:	48	(45.7%)	24	(22.9%)	
∘ Less than 10 cigarettes per day	11	(10.5%)	16	(15.2%)	**<0.001**
∘ 10–20 cigarettes per day	32	(30.5%)	8	(7.6%)	
∘ Over 20 cigarettes per day	5	(4.8%)	3	(2.9%)	
Physical activity level (scale, scoring 1–5)	3.0	(2.0–3.0)	4.0	(4.0–5.0)	**<0.001**
• 1. Very inactive *n* (%)	10	(9.5%)	3	(2.9%)	**<0.001**
• 2. Moderately inactive *n* (%)	26	(24.8%)	2	(1.9%)	
• 3. Nor active nor inactive *n* (%)	45	(42.9%)	18	(17.1%)	
• 4. Moderately active *n* (%)	20	(19.0%)	49	(46.7%)	
• 5. Highly active *n* (%)	4	(3.8%)	33	(31.4%)	
BMI (kg/m^2^)	25.7	(23.4–27.7)	26.7	(24.1–28.9)	0.057
• Underweight *n* (%)	1	(1.0%)	0	(0.0%)	0.278
• Normal-weight *n* (%)	44	(41.9%)	33	(31.4%)	
• Overweight *n* (%)	43	(41.0%)	54	(51.4%)	
• Obese *n* (%)	17	(16.2%)	18	(17.1%)	
Systolic arterial pressure	130	(120–140)	120	(120–130)	**<0.001**
Diastolic arterial pressure	80	(80–90)	80	(80–90)	0.438
Bone fractures before (any type of fractures):					**<0.001**
• No *n* (%)	35	(33.3%)	88	(83.8%)	
• Yes *n* (%)	70	(66.7%)	17	(16.2%)	
Season of examination/blood sampling:					0.318
• Late summer/early autumn *n* (%)	71	(61.0%)	64	(67.6%)	
• Late autumn/early winter *n* (%)	34	(39.0%)	41	(32.4%)	

BMI = body mass index, IQR = interquartile range, *p* = statistical significance of difference (bolded values are statistically significant, *p* < 0.05).

**Table 2 nutrients-16-02702-t002:** Comparison of dietary energy and macronutrient intakes between elderly subjects (>65 years) with and without fractures in the Vojvodina region, Serbia.

	With Fractures (*n* = 105)		Controls (*n* = 105)		
	Median	(IQR)	Median	(IQR)	*p*
Energy intake (Kcal/day)	1298.0	(1176.6–1531.7)	1736.5	(1491.9–1998.0)	**<0.001**
Protein intake (g/day)	38.6	(32.1–55.5)	70.3	(55.8–86.0)	**<0.001**
Fat intake (g/day)	55.6	(41.8–71.7)	82.1	(69.6–102.1)	**<0.001**
Carbohydrate intake (g/day)	157.8	(134.6–187.0)	159.9	(136.5–200.5)	0.284
Alcohol intake (g/day)	0.001	(0.000–0.005)	0.000	(0.000–0.003)	**0.049**
Fiber intake (g/day)	11.7	(9.4–14.8)	13.5	(10.5–18.0)	**0.003**
Energy intake from protein (%Kcal/day)	12.3	(10.1–17.0)	16.4	(14.5–18.9)	**<0.001**
Energy intake from fat (%Kcal/day)	38.3	(31.9–43.8)	44.4	(40.0–48.6)	**<0.001**
Energy intake from carbohydrate (%Kcal/day)	48.4	(40.4–54.9)	38.9	(34.1–43.4)	**<0.001**
Energy intake from alcohol (%Kcal/day)	0.000	(0.000–0.003)	0.000	(0.000–0.002)	**0.035**

IQR = interquartile range, *p* = statistical significance of difference (bolded values are statistically significant, *p* < 0.05).

**Table 3 nutrients-16-02702-t003:** Comparison of energy intakes through different food groups between elderly subjects (>65 years) with and without fractures in the Vojvodina region, Serbia.

	With Fractures (*n* = 105)		Controls (*n* = 105)		
	Median	(IQR)	Median	(IQR)	*p*
Energy intake from milk and milk products (Kcal/day)	87.9	(42.1–166.1)	222.7	(111.7–331.0)	**<0.001**
Energy intake from eggs and egg products (Kcal/day)	11.5	(0.0–51.7)	64.5	(14.5–131.4)	**<0.001**
Energy intake from meat and meat products (Kcal/day)	122.3	(68.9–224.7)	160.9	(75.8–292.7)	0.181
Energy intake from fish, seafood, and related products (Kcal/day)	0.0	(0.0–0.0)	93.1	(0.0–179.0)	**<0.001**
Energy intake from edible fats, oil, and similar products (Kcal/day)	168.6	(107.3–236.2)	249.5	(167.2–359.9)	**<0.001**
Energy intake from grains and grain products (Kcal/day)	482.2	(367.0–600.5)	535.0	(383.8–680.9)	0.073
Energy intake from nuts, seeds, and related products (Kcal/day)	21.7	(1.2–51.3)	20.4	(3.1–52.4)	0.580
Energy intake from vegetables and vegetable products (Kcal/day)	71.2	(34.0–119.8)	69.6	(44.5–123.5)	0.512
Energy intake from fruit and fruit products (Kcal/day)	44.0	(0.0–98.2)	56.2	(3.3–116.7)	0.067
Energy intake from sugar and sugar products (Kcal/day)	62.9	(9.3–154.9)	22.9	(0.0–69.5)	**<0.001**
Energy intake from alcoholic and non-alcoholic beverages (Kcal/day)	35.8	(1.3–86.7)	32.2	(1.7–62.2)	0.151
Energy intake from miscellaneous food products (Kcal/day)	0.6	(0.0–1.4)	1.4	(0.3–2.6)	**<0.001**
Energy intake from special nutritional supplements (Kcal/day)	0.0	(0.0–0.0)	0.0	(0.0–0.0)	0.317

IQR = interquartile range, *p* = statistical significance of difference (bolded values are statistically significant, *p* < 0.05).

**Table 4 nutrients-16-02702-t004:** Logistic regression for predicting the risk for fractures in the whole sample (*n* = 210).

Model No	Predictors	B	Exponent (B) (Odds Ratio)	95%CI for Exponent (B)	Predictor Significance *p*	Model Nagelkerke R^2^	Model Significance *p*
**Model 1**	Constant	5.951	383.986		**<0.001**	0.755	**<0.001**
	Serum 25(OH)D levels (nmol/L)	−0.123	0.884	(0.854–0.915)	**<0.001**		
**Model 2**	Constant	16.047	9,309,443.280		**0.004**	0.817	**<0.001**
	Serum 25(OH)D levels (nmol/L)	−0.131	0.878	(0.841–0.916)	**<0.001**		
	Physical activity level (1–5)	−0.735	0.480	(0.284–0.811)	**0.006**		
	Season (late autumn/early winter)	1.012	2.752	(0.882–8.584)	0.081		
	BMI (kg/m^2^)	−0.105	0.900	(0.766–1.057)	0.200		
	Smoking status (1–5)	0.244	1.276	(0.864–1.886)	0.221		
	Age (years)	−0.069	0.933	(0.832–1.046)	0.234		
	Sex (male)	−0.395	0.674	(0.169–2.680)	0.575		
**Model 3**	Constant	9.975	21,485.246		**<0.001**	0.863	**<0.001**
	Serum 25(OH)D levels (nmol/L)	−0.146	0.864	(0.822–0.909)	**<0.001**		
	Protein intake (g/day)	−0.062	0.939	(0.907–0.973)	**<0.001**		
	Season (late autumn/early winter)	2.554	12.858	(2.933–56.373)	**0.001**		
	Vitamin D intake (µg/day)	−0.111	0.895	(0.796–1.006)	0.064		
**Model 4**	Constant	14.339	1,688,174.056		**<0.001**	0.874	**<0.001**
	Serum 25(OH)D levels (nmol/L)	−0.155	0.857	(0.808–0.909)	**<0.001**		
	Season (late autumn/early winter)	2.240	9.393	(2.196–40.174)	**0.003**		
	Fiber intake (g/day)	−0.173	0.841	(0.739–0.958)	**0.009**		
	Fat intake (g/day)	−0.034	0.966	(0.941–0.992)	**0.012**		
	Physical activity level (1–5)	−0.733	0.481	(0.252–0.916)	**0.026**		
	Vitamin D intake (µg/day)	−0.105	0.900	(0.800–1.012)	0.079		
**Model 5**	Constant	14.708	2,440,269.079		**<0.001**	0.919	**<0.001**
	Serum 25(OH)D levels (nmol/L)	−0.231	0.794	(0.713–0.885)	**<0.001**		
	Fish and seafood dietary intakes (Kcal/day)	−0.028	0.972	(0.958–0.987)	**<0.001**		
	Season (late autumn/early winter)	3.817	45.472	(3.799–544.322)	**0.003**		
	Egg and egg product dietary intakes (Kcal/day)	−0.018	0.982	(0.969–0.996)	**0.011**		
	Sugar and sweets dietary intakes (Kcal/day)	0.013	1.013	(1.002–1.025)	**0.022**		
	Physical activity level (1–5)	−0.842	0.431	(0.199–0.933)	**0.033**		
	Milk and dairy product dietary intakes (Kcal/day)	−0.008	0.992	(0.985–0.999)	**0.034**		

**Model 1**, univariate: non-adjusted, only vitamin D serum levels as a predictor variable; **Model 2**, multivariate, variable selection method—enter: vitamin D serum levels as a predictor variable, adjusted for sex, age, BMI, season, smoking status, and physical activity level; **Model 3**, multivariate, variable selection method—stepwise (forward, conditional): extended, with predictor variables: vitamin D serum levels, vitamin D dietary intakes, vitamin D supplements intake, dietary calcium intakes, dietary protein intakes, dietary fat intakes, dietary fiber intakes, adjusted for sex, age, BMI, season, smoking status, and physical activity level; **Model 4**, multivariate, variable selection method—stepwise (forward, conditional): extended, with predictor variables: vitamin D serum levels, vitamin D dietary intakes, vitamin D supplements intake, dietary calcium intakes, dietary fat intakes, dietary fiber intakes, adjusted for sex, age, BMI, season, smoking status, and physical activity level; **Model 5**, multivariate, variable selection method—stepwise (forward, conditional): extended, with predictor variables: vitamin D serum levels, vitamin D dietary intakes, vitamin D supplements intake, dietary calcium intakes, dietary milk and dairy product intakes, dietary egg and egg product intakes, dietary fish and seafood intakes, dietary edible fats and oils intake, adjusted for sex, age, BMI, season, smoking status, and physical activity level; BMI = body mass index, B = logistic regression coefficient, CI = confidence interval, R^2^ = coefficient of determination, *p* = statistical significance of difference (bolded values are statistically significant, *p* < 0.05).

## Data Availability

Dataset available on request from the authors.

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
