# Peer review of "Impact of Vitamin D Status and Nutrition on the Occurrence of Long Bone Fractures Due to Falls in Elderly Subjects in the Vojvodina Region of Serbia"

_nutrients, 2024, doi:10.3390/nu16162702_

Round 1

Reviewer 1 Report

Comments and Suggestions for Authors

This outstanding article examines the somewhat controversial link between vitamin D status, vitamin D supplementation, and bone fractures. The paper is well elaborated, along with informative tables and figures, guided by sound statistics. I have some comments:

Methods: have all the fractures in the study been directly linked to osteoporosis? How did you confirm that?

Discussion: while there exists a clear link between vitamin D-levels and osteoporosis-based fractures, it has never been proven that oral supplementation could anyhow alleviate that risk (incl. several meta-analyses). As such, your results - see also table 4 - appear to contradict the existing literature. Do you have an explanation for this interesting feature?

Author Response

Thank you very much for taking the time to review this manuscript, very positive evaluation, and insightful and useful comments.

Please find the detailed responses below to both reviewers and the corresponding revisions/corrections highlighted in yellow in the re-submitted files.

Reviewer 2 Report

Comments and Suggestions for Authors

This paper paper is well written and examines the impact of diet and vitamin D status on the risk of long bone fractures due to falls in 19 elderly subjects in Vojvodina, Serbia

In order to improve the paper please:

1) Clarify the study design and participant selection by providing more details on how the control group was matched to the fracture group in terms of age, sex, BMI, and other key characteristics

2) Expand the discussion on the role of vitamin D in bone health and fracture risk by citing relevant studies that support the findings and provide context.

3)Analyze the dietary intake data in more detail, such as comparing the percentage of energy from macronutrients and the adequacy of micronutrient intakes compared to recommendations.

4) Assess the potential confounding effects of physical activity and other lifestyle factors on the association between diet, vitamin D status, and fracture risk.

5) Discuss the limitations of the study, such as the cross-sectional design, self-reported dietary intake, and potential selection bias in the control group.

6) Provide more information on the statistical methods used, such as the criteria for inclusion in the logistic regression models and the handling of missing data

Author Response

(The authors gave the same response as above.)
